# Fabrication and Characterisation of Calcium Sulphate Hemihydrate Enhanced with Zn- or B-Doped Hydroxyapatite Nanoparticles for Hard Tissue Restoration

**DOI:** 10.3390/nano13152219

**Published:** 2023-07-31

**Authors:** Adrian Ionut Nicoara, Teodor Gabriel Voineagu, Andrada Elena Alecu, Bogdan Stefan Vasile, Ioana Maior, Anca Cojocaru, Roxana Trusca, Roxana Cristina Popescu

**Affiliations:** 1Faculty of Chemical Engineering and Biotechnologies, University Politehnica of Bucharest, 060042 Bucharest, Romania; adrian.nicoara@upb.ro (A.I.N.); andrada.alecu16@gmail.com (A.E.A.); ioana.maior@upb.ro (I.M.); truscaroxana@yahoo.com (R.T.); 2National Research Center for Micro and Nanomaterials, University Politehnica of Bucharest, 060042 Bucharest, Romania; bogdan.vasile@upb.ro; 3National R&D Institute for Nonferrous and Rare Metals–IMNR, 077145 Bucharest, Romania; 4Faculty of Medical Engineering, University Politehnica of Bucharest, 060042 Bucharest, Romania; teodor.voineagu@stud.fim.upb.ro (T.G.V.); roxana.popescu@upb.ro (R.C.P.); 5Research Center for Advanced Materials, Products and Processes, University Politehnica of Bucharest, 060042 Bucharest, Romania; 6National R&D Institute for Physics and Nuclear Engineering-Horia Hulubei, 077125 Magurele, Romania

**Keywords:** nanomaterials, bone reparation, gypsum, doped hydroxyapatite, bone cement

## Abstract

A composite based on calcium sulphate hemihydrate enhanced with Zn- or B-doped hydroxyapatite nanoparticles was fabricated and evaluated for bone graft applications. The investigations of their structural and morphological properties were performed by X-ray diffraction (XRD), Fourier transform infrared (FTIR) spectroscopy, scanning electron microscopy (SEM), and energy dispersive X-ray (EDX) spectroscopy techniques. To study the bioactive properties of the obtained composites, soaking tests in simulated body fluid (SBF) were performed. The results showed that the addition of 2% Zn results in an increase of 2.27% in crystallinity, while the addition of boron causes an increase of 5.61% compared to the undoped HAp sample. The crystallite size was found to be 10.69 ± 1.59 nm for HAp@B, and in the case of HAp@Zn, the size reaches 16.63 ± 1.83 nm, compared to HAp, whose crystallite size value was 19.44 ± 3.13 nm. The mechanical resistance of the samples doped with zinc was the highest and decreased by about 6% after immersion in SBF. Mixing HAp nanoparticles with gypsum improved cell viability compared to HAp for all concentrations (except for 200 µg/mL). Cell density decreased with increasing nanoparticle concentration, compared to gypsum, where the cell density was not significantly affected. The degree of cellular differentiation of osteoblast-type cells was more accentuated in the case of samples treated with G+HAp@B nanoparticles compared to HAp@B. Cell viability in these samples decreased inversely proportionally to the concentration of administered nanoparticles. From the point of view of cell density, this confirmed the quantitative data.

## 1. Introduction

Bones are the major part of the musculoskeletal system that supports body weight, performs motion, and protects the internal organs [1]. They have functional adaptation ability, meaning that bones can adjust their mass and architecture according to their mechanical environment or other mechanical stimuli. Bone grafting is one of the most commonly used strategies for treating bone defects and is widely applied for bone regeneration in orthopaedic surgeries. An ideal bone graft must augment the process of bone healing with optimal stability and durability, along with characteristics including osteogenesis, osteoinduction, and osteoconduction [2].

Bone regeneration, or fracture healing, is a fascinating process in which bone is able to self-regenerate and reach its initial function without leaving any types of scar tissue [3]. Despite its potential for robust healing, there are conditions in which bone fails to heal, leading to delayed or systemic bone loss (osteoporosis and osteopenia) and large bone defects due to trauma or cancer. There are two fundamental physiological processes of bone functional adaptation: modelling and remodelling. In the modelling process, continuous bone resorption and formation occur separately at different locations, changing the whole bone morphology [4]. Remodelling occurs in all in vivo bone tissues and is an important way to renew bone.

Calcium sulphate (CS) is the simplest bone substitute. It has been widely recognised as a well-tolerated and readily available material with the most affordable price and a prolonged history of clinical use for grafting [5]. Calcium sulphate benefits from a crystalline structure and biodegradation to support space for cell growth and increase extracellular calcium ions [6,7]. On the other hand, calcium sulphate cements have also been investigated as alternative candidates to autograft in restoring bone defects [5,8,9,10]. Gypsum mainly consists of calcium sulphate dihydrate, a promising raw material for producing calcium sulphate hemihydrate. When gypsum is heated to over 110 °C, water can be removed in a process known as calcination, resulting in calcium sulphate hemihydrate (CSH) production. CSH is well known for its suitable clinical applications, such as bone void filler, since it possesses key features including osteoconductivity, excellent workability, high self-setting strength, biocompatibility, rapid and complete resorption with minimal inflammation, and is relatively cost-effective. Calcium sulphate cements, especially calcium sulphate dihydrate (CSD—CaSO_4_·2H_2_O) and the derivative of calcium sulphate hemihydrate (CSH—CaSO_4_·0.5H_2_O) after the mixing of the powder CSH with water, have been extensively used to fill bone defects due to both their capability for bone repair and their excellent biocompatibility. CSH [11], used in this study, continues to be the object of research and interest as one of the most successful bone cements because it has (i) the ability to undergo in situ setting after filling the defects [12], (ii) good biocompatibility without inducing an inflammatory response [13], and (iii) promotes bone healing [11,14,15].

Hydroxyapatite (Hap-Ca_10_(PO_4_)_6_(OH)_2_) is a calcium phosphate mineral found in bones. HAp has been recognised for orthopaedic implant coating, drug delivery, graft replacement, as a constituent of bone scaffolds, and bone defect filler application due to its osteoinduction, osteoconduction, osteointegration, and bioactive potential [16,17]. The biological and mechanical properties of HAp are affected by its composition, structure, size, and morphology.

Hydroxyapatite-based biomaterials are among the most widely used materials in bone regeneration. They are biocompatible and bioactive and mimic the chemical and structural properties of natural bone. Hydroxyapatite is a calcium phosphate mineral found in natural bone and is the major component of the mineral phase of the bone tissue. Hydroxyapatite-based biomaterials have been extensively studied for their ability to support bone regeneration and have been used in a variety of medical applications, including dental implants, bone grafts, and orthopaedic implants. Hydroxyapatite-based biomaterials are also versatile and can be used in a variety of forms, including powders, coatings, and porous structures [18]. This allows for different applications depending on the specific needs of the patient. For example, hydroxyapatite coatings can be used to improve the integration of metal implants with bone tissue, while porous hydroxyapatite structures can be used as bone graft substitutes [19,20,21]. Despite the many advantages of hydroxyapatite-based biomaterials, there are also some limitations. One challenge is achieving the right balance of porosity and mechanical strength. While a highly porous scaffold is ideal for supporting bone growth, it may not be strong enough to withstand the forces placed on it during normal activity. On the other hand, a too-dense scaffold may not allow for adequate bone growth [22].

Since the selective substitution of its cations (Ca^2+^) and anions (OH^−^ and/or PO_4_^3−^) is allowed in HAp, it can become an ideal host for ionic doping [23]. Because HAp is weak in osteoinduction, ion doping can improve its biological activities. Different dopants have been studied, and research is ongoing to tailor synthetic HAp for different medical applications [24]. Zinc is an essential mineral that is also the second-most abundant mineral in the human body. Zinc deficiency can affect the central nervous, skeletal, and reproductive systems, as well as physical growth, and increase the risk of infection. Hydroxyapatite doped with zinc has been extensively studied in the last few years. Uysal et al. [24] published an extensive review of the different synthesis methods and sintering parameters for doped hydroxyapatite published during the last 20 years [18,25,26,27,28,29,30]. Negrila et al. [25] and Predoi et al. [26] used sol-gel to produce HAp doped with Zn^2+^. Predoi et al. [26] also analysed the influence of the stability of Zn-HAp solutions on antibacterial properties. They found that Zn content has a significant impact on solution stability and prevents bacterial colonisation. Begam et al. [27] found that Zn^2+^ doping changes the lattice parameter of HAp, increasing cell adhesion and growth.

Boron is an important trace element for plants, but it is not as important for animals. However, boron has been shown to play a role in osteoblastic driving [31]. Tunçay et al. [32] first attempted microwave-assisted biomimetic precipitation of B-HAp and showed that B-HAp accelerates cell attachment and differentiation and facilitates early mineralisation.

The present study is focused on the synthesis of composites prepared by mixing CSH with Zn- or B-doped hydroxyapatite and the investigation of their structural and morphological properties by X-ray diffraction (XRD), Fourier transform infrared (FTIR) spectroscopy, scanning electron microscopy (SEM), and energy dispersive X-ray (EDX) spectroscopy techniques. The results provided from this study could lead to the production of bone cement that could be used as complementary materials for filling bone defects and for their effective healing. To study the bioactive properties of the obtained composites, soaking tests in simulated body fluid (SBF) were performed. Moreover, the biological behaviour and cell proliferation of osteoblast-like MG-63 cells were investigated following incubation with nanostructured CSH-zinc/boron-doped hydroxyapatite composites. 

## 2. Materials and Methods

### 2.1. Composite Preparation

All samples based on synthetic hydroxyapatite were obtained starting with the following precursors: Ca(NO_3_)_2_·2H_2_O (Sigma-Aldrich, St. Louis, MO, USA, ≥98.0%), (NH_4_)_2_HPO_4_ (Sigma-Aldrich, St. Louis, MO, USA, reagent grade, ≥98.0%), H_3_BO_3_ (Sigma-Aldrich, St. Louis, MO, USA, ≥99.5%), and Zn(NO_3_)_2_·2H_2_O (Sigma-Aldrich, St. Louis, MO, USA, reagent grade, 98%). Calcium nitrate solution (0.167 mol L^−1^) was mixed with diammonium hydrogen phosphate solution (0.1 mol L^−1^). For the doped hydroxyapatite, the required amount of zinc nitrate or boric acid precursors was added to obtain doping of 2 wt.% for Zn and 5 wt.% for B.

After mixing, the pH of the solution was measured (Table 1) and adjusted to 10.5 with the addition of ammonium hydroxide (NH_4_OH). Afterward, the mixture was subjected to a hydrothermal treatment for 12 min at 120 °C, washed with distilled water, and dried at 60 °C for 24 h. 

To obtain semi-hydrate gypsum powder, calcium sulphate dihydrate was subjected to hydrothermal treatment at 132 °C for 30 min and left overnight at 50 °C in the oven.

The composites were obtained by mixing calcium sulphate with the previously obtained hydroxyapatites. The composition and notations of the obtained samples are presented in Table 2.

The composite was obtained by mixing undoped or doped hydroxyapatite with gypsum (1:1 weight ratio) and a 10% glycerol aqueous solution. The amount of solution used for each mixture was dosed judiciously to keep the same paste consistency.

The necessary volume of liquid was related to the size of the powders obtained. The doped hydroxyapatite showed smaller particles. The obtained mixture was poured into a cylindrical mould (ø = 10 mm and h = 10 mm) and characterised after hardening.

### 2.2. Samples Characterisation

The X-ray diffraction (XRD) technique was used to determine the degree of crystallinity, the crystallite size, and the phases present in the samples. The analysis was carried out using a PANalytical Empyrean diffractometer (Almelo, Netherlands) at room temperature with a characteristic Cu X-ray tube (λCuKα1= 1.541874 Å). The samples were scanned in a Bragg–Brentano geometry with a scan step increment of 0.02 ° and a counting time of 100 s/step. The XRD patterns were recorded in the 2θ angle range of 5–80°. Rietveld quantitative phase analysis was performed using the X′Pert High Score Plus 3.0 software (PANalytical, Almelo, The Netherlands). After refining, values were obtained between 1.44% and 1.78% for goodness of fit, 6.65% and 7.08% for Rexpected, and 6.67% and 7.59% for Rprofile. The crystallite size was determined by Debye–Scherer Equation (1)
(1)$$\sigma =\frac{k\lambda }{\beta \cos\theta }\;(\mathrm{nm})$$
where s = crystallite size (nm), k = the Scherrer constant (0.98), λ = denotes the wavelength (0.154 nm), and β = the full width at half maximum (FWHM) in radians

Morphological aspects were studied via scanning electron microscopy (SEM) with a Quanta Inspect F50 microscope coupled with an energy dispersive spectrometer (EDS) and a Titan Themis 200 transmission electron microscope (TEM) with a line resolution of 90 pm in high-resolution transmission electron microscopy (HRTEM) mode (Thermo Fisher, Eindhoven, The Netherlands). The mechanical compression strength was determined using the Shimadzu Autograph AGS-X 20kN (Shimadzu, Tokyo, Japan) equipment. Fourier transform infrared spectroscopy (FTIR) investigation was performed using a Nicolet iS50R spectrometer (Thermo Fisher, Waltham, MA, USA). The measurements were made at room temperature using the total reflection attenuation module. Each sample was scanned 32 times between 4000 and 400 cm^−1^, at a resolution of 4 cm^−1^. 

Raman spectra were collected using a LabRam HR Evolution HORIBA (Palaiseau, France), acquisition time 2 s, accumulation 20, laser 514 nm, hole diameter 100 microns, objective 50×, grating 600 g/mm, ND filter 100%, range 300–1100 cm^−1^, with a measurement error of ±0.5 cm^−1^.

In order to evaluate the stability in a wet environment [33], the cylinders obtained from each composite were placed in 20 mL of simulated body fluid with pH = 7.4 (SBF) prepared according to the Kokubo recipe [34]. The initial mass was recorded, and the samples were immersed in SBF solution. After a certain amount of immersion, samples were removed, dried, and weighed again. The mass loss could be calculated using Equation (2):(2)$$\mathrm{Weight}\;\mathrm{loss}\;=\frac{w_{i\;}-{\;w}_{f}}{w_{i}}\;100\;(\%)$$
where w_i_ = sample weight before immersion and w_t_ = dried sample weight after t min of immersion in SBF.

The mechanical compressive strength was determined by pressing the samples until the breaking point with a speed of 1 mm/min using Shimadzu Autograph AGS-X 20kN equipment (Shimadzu, Tokyo, Japan). The test was made in triplicate on samples hardened for 3, 7, and 28 days, according to the standard [35]. In addition to the initial composites (hardened in the air), samples immersed for 72 h in SBF were also tested.

The in vitro biological behaviour was investigated using osteoblast-like MG-63 cells (CLS, Heidelberg, Germany) that were cultured in Eagle’s Minimum Essential Medium (EMEM, Gibco, Billings, MT, United States), supplemented with 10% foetal bovine serum (FBS, Gibco, Billings, MT, United States), under standard conditions of temperature and humidity (37 °C, 5% CO_2_, 90% humidity). 

The hydroxyapatite nanoparticles were suspended in deionised water at a concentration of 0.012 g/mL by dispersing with an ultrasound probe and then sterilised by gamma radiation. MG-63 cells were seeded in 96-well plates at a concentration of 2000 cells/well and incubated under standard conditions to allow cell attachment. After 24 h, the culture medium was removed and replaced with culture medium with nanoparticles at different concentrations (0, 25, 50, 100, and 200 µg/mL, previously prepared by dispersing in complete culture medium with an ultrasound bath). The cells were then incubated in the presence of hydroxyapatite nanoparticles for 7 days under standard temperature and humidity conditions.

Following incubation in the presence of nanoparticles, investigations related to cell morphology, viability, and cell differentiation were carried out. Investigations of cell viability and proliferation were performed using the MTT tetrazolium salts assay (3-(4,5-dimethylthiazol-2-yl)-2,5-diphenyltetrazolium bromide) [36]. Seven days after treatment, the nanoparticle solution was removed and replaced with 10% MTT (5 mg/mL) in cell culture medium. After 2 h incubation, the reacted formazan crystals were solubilised using DMSO, and the absorbance of the supernatant was measured at 570 nm. 

Cell differentiation was tested using Alizarin red assay [37]. Following incubation, the nanoparticle-containing cell culture medium was removed, and cells were fixed using 4% paraformaldehyde in PBS for 1 h. After this, the supernatant was removed, and cells were washed with PBS several times to remove any residues of nanoparticles left. Following this step, 40 mM Alizarin red was added to each well and incubated for 45 min at room temperature. Then, the supernatant was removed, and cells were thoroughly washed with deionised water. This method is based on the specific staining of Ca deposits in the cells resulting from osteoblast cell differentiation. The stained Ca deposits were then dissolved in 10% acetic acid, and the supernatant was collected and heated at 85 °C for 10 min. Following centrifugation at 20,000× *g* for 10 min, the supernatant was neutralised with 10% ammonium hydroxide. The absorbance was measured at 405 nm.

Experiments were performed in triplicate, and cell viability and differentiation ability, respectively, were calculated by relating the data obtained for each sample to the negative control samples (for which a value of 100% was assigned). Data were expressed as ± SEM (standard error of the mean), and their statistical evaluation was performed using the Student t-test function, for which * *p* < 0.05, ** *p* < 0.01, and *** *p* < 0.001. There were blank samples (nanoparticles without cells at the investigated concentrations) for all quantitative determinations, whose absorbance was subtracted from that of cellular samples.

## 3. Results

### 3.1. HAp, HAp@Zn and HAp@B Powder Characterisation

The XRD profiles of HAp, HAp@B, and HAp@Zn prepared powders are displayed in Figure 1. The XRD pattern of HAp shows a single crystalline phase identified through the characteristic diffraction maxima. The phase identification is performed using the PDF 04-008-4763 file. Miller indices corresponding to the position of the crystallographic planes and directions could be associated with each important peak, such as (002), (121), (030), (222), and (123). The X-ray diffractogram indicates the presence of two high-intensity peaks located near 26° and 32°, which are in good agreement with the literature data for HAp [38,39].

Figure 1b shows the magnification of the area with the most intense peaks for the studied samples. The diffractograms for the HAp@Zn and HAp@B samples are similar to those for the HAp sample. However, the diffraction peaks are smaller in intensity, with the maximum being shifted and broader, which, according to Miyaji et al. [40], suggests that the crystallinity of apatite increases in the presence of a dopant. 

In order to determine the degree of crystallinity, the average crystallite size, the structural microstrains, and the volume of the elementary cell, the Rietveld analysis was performed. The results of the refining are presented in Table 3.

As can be seen, following the Rietveld analysis, the addition of the dopant tends to determine an increase in the crystallinity of powders. The addition of 2% Zn increases crystallinity by approximately 2.27%, while the addition of B causes an increase of approximately 5.61% compared to the undoped HAp sample. Also, the crystallite size decreases as a result of doping. In the case of B, a crystallite size of 10.69 ± 1.59 nm is recorded, and in the case of Zn doping, the size value reaches 16.63 ± 1.83 nm, while undoped HAp has a crystallite size value of 19.44 ± 3.13 nm.

Considering the atomic radius OF the dopants (Zn = 135 pm and B = 85 pm), their addition to the crystalline network of hydroxyapatite determines the change in the volume of the elemental cell and the microstrains of the structure. Since B has an atomic radius closer to that of P (100 pm), it is assumed to prefer its positions in the network and not the Ca (180 pm) positions. Therefore, a stronger tension caused by crystal defects present in the network is observed, as proven by the microstrain value of 0.86% compared to undoped HAp, which shows a value of 0.47%.

In order to complete the information from XRD studies related to the structure of the samples, Raman spectroscopy investigations were performed. The obtained spectra are presented in Figure 2.

Following Raman spectroscopy, it is possible to observe the bands located at 431 and 445 cm^−1^, which can be attributed to the bending vibration of the O-P-O bonds doubly degenerated (ν2). The bands located at values of 578 and 590 cm^−1^ were attributed to the bending vibration of the O-P-O bonds, which triply degenerated (ν4). The intense band located at 961 cm^−1^ was attributed to the stretching of the non-degenerate P-O symmetric bond (ν1). Additionally, the bands located at values of 1046 and 1075 cm^−1^ were attributed to the antisymmetric stretching of the P-O bond triple degenerated (ν3) [41,42]. 

Analysing the band located at 961 cm^−1^, characteristic of the vibration of PO_4_ tetrahedra, it is observed that it widens with the addition of the dopant. The ratio between the intensities of the ν1 and ν2 vibration modes decreases from 3.59 for HAp to 2.98 for HAp@Zn, respectively, to 2.48 for HAp@B, which indicates the distortion and increase in the disorder degree of the crystalline network [43].

The shape and size of the particle aggregates were determined by SEM. The images obtained for HAp, HAp@Zn, and HAp@B are presented in Figure 3 at different magnifications.

In the case of the HAp, medium-sized particles are observed around the value of 67.11 nm, while the values decrease with the addition of dopants, reaching 54.67 nm for the sample doped with B and 52.64 nm for the sample doped with Zn. Also, due to the small size and the pseudo-acicular shape, the samples show an accentuated agglomeration tendency.

The presence of characteristic hydroxyapatite elements (Ca, P) and dopants was demonstrated by the EDS spectra and the elemental composition of the powder presented in Table 4.

Boron was found in HAp particles in a proportion of 3.72 wt.% (Table 4), while Zn was found in a proportion of 0.33 wt.%. The presence of carbon can be attributed to sample preparation for SEM analysis. However, it can also be attributed to the possible carbonation of the HAp particles under the influence of the synthesis conditions and the CO_2_ present in the atmosphere [25].

The TEM images obtained on the HAp samples highlight well-defined particles with a polyhedral, irregular shape, with sizes between 30–70 nm for the HAp (Figure 4a), 15–75 nm for the HAp@Zn sample, and 40–65 nm for the HAp@B sample (Figure 4d). Also, in the case of doping with Zn, the particles are presented in acicular hexagonal form (Figure 4g). The polycrystalline character of all samples is highlighted by SAED analysis (Figure 4c,f,i).

Performing the measurements on the HRTEM images (Figure 4b,e,h), it was found that the distances between the atomic chains vary between 3.5 Å for the HAp sample, 3.4 Å for HAp@Zn, and 3.3 Å for HAp@B. 

The mapping of the main elements on the sample surface is presented in Figure 5.

As shown in Figure 5, in addition to the characteristic elements of hydroxyapatite (Ca, P, and O), the presence of carbon could also be identified in the sample. This can be attributed to superficial carbonation of the sample, most likely during synthesis following interaction with atmospheric CO_2_. The elemental distribution of the dopants indicates that they are distributed uniformly among the powder particles.

### 3.2. Gypsum Analysis

Gypsum is a biomaterial that has been used as bone cement for many years [44]. It is a versatile material used in several medical applications due to its biocompatibility, biodegradability, and easy availability. Gypsum is a naturally occurring mineral made up of calcium sulfate. It is widely used in construction and medical applications, such as making casts and moulds. Gypsum can also be easily modified to adjust its properties, which makes it an excellent candidate for use in bone cements that must meet specific mechanical and biological requirements. When used as bone cement, gypsum is mixed with water to create a paste-like substance that can be moulded to the shape of the bone. The paste then hardens to create a rigid bond, much like traditional cement. Gypsum can also be reinforced with other materials, such as carbon fibers, to increase its strength and durability.

Figure 6 shows the X-ray diffractogram of the gypsum powder. The specific peaks of calcium sulphate hemihydrate are observed, with the most prominent peak at an angle of 11.7°, according to the PDF 04-015-7420 file. Also, each important peak has Miller indices associated with it, corresponding to the plane position and crystallographic direction associated with the peak.

After carrying out the FTIR analysis, the specific graph indicates the structural composition of calcium sulphate. Thus, one can observe the vibration bands characteristic of stretching the O-H bond at values 3200–3600 cm^−1^, but also between 1500–1700 cm^−1^, and the sulphate bond (S-O) characteristic bands of between 850–1230 and also 400–800 cm^−1^ [45,46].

The EDS analysis performed on the plaster sample shows specific chemical elements such as oxygen (O) at 0.53 keV, sulphur (S) at 2.31 keV, and calcium (Ca) at a value of 3.7 keV. The percentage composition of the elements is presented in Figure 7.

### 3.3. Composites Material Characterisation

After hardening, the composites, obtained by mixing the gypsum with the HAp-type particles and the aqueous solution, were matured for 28 days and characterised by FTIR. The results are presented in Figure 8.

The characteristic bands of the groups found in CaSO_4_·2H_2_O are present in the wave range 3300–3600 cm^−1^. The bands characteristic of the P-O bond are located between 2775–3050 cm^−1^, and those characteristic of hydration water are located around 1650–1720 cm^−1^. The bands attributed to the C-O bonds from glycerol are located between 1580–1635 cm^−1^ and between 1370–1520 cm^−1^, and the PO_4_^−3^ group between 900–1220 cm^−1^ and between 400–740 cm^−1^.

Scanning electron microscopy was performed on each composite sample, and the results are shown in Figure 9.

The morphology of the composite samples is similar and is predominantly formed by the characteristic forms of calcium sulphate dihydrate. Samples are formed by crystals with sizes between 200–350 nm. As shown in Figure 9, hydroxyapatite does not significantly influence the behaviour of the samples; HAp particles are uniformly distributed on the surface of the gypsum crystals as a result of the mechanical homogenisation process. At low magnifications, one can see the porosity obtained by the intercalation of these polyhedral crystals.

To perform in vitro tests, the composites were moulded into cylindrical shapes and, after hardening, immersed in SBF solution. The weight loss variation of composite samples G+HAp, G+HAp@B, and G+HAp@Zn during 72 h of immersion in SBF is presented in Figure 10.

The rate of degradation of the composites in the SBF environment is accelerated in the first 15 h when a decrease in the weight of the samples of approximately 37% is recorded. After 15 h, a decrease in degradation rate is observed, with the sample reaching a maximum of approximately 47% after 72 h of contact with the liquid. This rate of disintegration of the resulting paste may coincide with the beginning of cell proliferation to form new bone tissue.

It is also noted that G+HAp@Zn samples tend to disintegrate more slowly in the presence of SBF than the other types of studied composites that showed similar behaviour. The visual aspects of the samples before and after immersion in SBF for 72 h are shown in Figure 11, and SEM images are presented in Figure 12.

Keeping the samples in SBF leads to a rearrangement of the microstructure. As can be seen in the SEM images, the gypsum crystals, initially formed following the hardening process, decrease in size and flatten due to the dissolution process. This process is followed by the mineralisation of the crystal surface through the deposition of apatite phases that lead to a more compact structure. This formation of new apatite phases is also confirmed by the FTIR analysis (Figure 13).

The bands detected at 566, 601, 962, 1039, and 1089 cm^−1^ belong to the phosphate phase [47,48]. The band corresponding to the hydroxyl group can be observed at 631 and 3550 cm^−1^ [49], and the band attributed to water molecules is observed around 3050–3550 cm^−1^. 

Also, specific bands for carbonate groups were detected at 872, 1421, and 1467 cm^−1,^ and the band of calcium sulphate is present at a value of 3380 cm^−1^ [50,51,52]. 

Compared to the FTIR performed on the composites before immersion in SBF (see Figure 8), a decrease in the characteristic bands for calcium sulphate and water and an increase in absorption for the PO_4_^3−^ bands are observed. This indicates the formation of a new apatite phase.

Regarding the mechanical resistance of the samples at various hardening intervals (Table 5), it was found that the most resistant was the sample doped with Zn. Also, the values for the samples immersed in SBF are 1.08 MPa for the G+HAp sample, 0.74 MPa for the G+HAp@B sample, and 1.93 MPa for the G+HAp@Zn sample. From these values, it can be seen that the samples gain strength for up to 28 days. After introducing samples in SBF, these resistances decrease by about 6%.

The biological evaluation of the gypsum/hydroxyapatite-modified nanoparticles was conducted for MG-63 osteoblast-like cells in terms of proliferation measurements and the ability of the cells to differentiate and form bone tissue following mineralisation. Osteoblast proliferation in the presence of nanovehicles is an important parameter to follow in terms of studying the stimulation abilities of osteoblasts. In this regard, the MTT tetrazolium-salt viability assay was performed following 7 days of incubation in the presence of nanoparticles. This method measures mitochondrial metabolism. The ability to metabolise the substance is proportional to cell viability. The analysis performed 7 days following cell seeding can also measure osteoblast cell proliferation in the presence of the nanoparticles because this incubation time is longer than one cell cycle. 

The results are comparatively shown in Figure 14, depending on the type of HAp. In the case of (G)/HAp, the cell metabolic activity decreased proportionally with the increase in nanoparticle concentration (*p* < 0.05 for 200 μg/mL). Gypsum alone proved to have a biocompatible behaviour at all concentrations, as the cell viability did not decrease below the 70% threshold for any of the investigated concentrations. With the addition of gypsum to the hydroxyapatite nanoparticles, cell viability was considerably improved compared to HAp alone. At all concentrations, unless the highest concentration was employed in the study (200 μg/mL), a reduction in cell metabolism was observed. For 200 μg/mL of G+HAp, a reduction in mitochondrial metabolism was observed compared to HAp. However, the difference noticed between these two samples was not statistically significant (G+HAp vs. HAp, NS). Overall, all samples proved biocompatible at concentrations below 100 μg/mL. A statistically significant proliferation of osteoblast-like cells was noticed in the case of G+HAp at 50 μg/mL (*p* < 0.01 compared to the negative control). 

The modification of HAp with B did not show any significant changes in the behaviour of the cells in terms of viability and proliferation compared to (gypsum)/HAp. The cells’ metabolic activity decreases with the increase in nanoparticle concentration. HAp@B showed a biocompatible behaviour at concentrations of 100 μg/mL and above, while 200 μg/mL induced a cytotoxic effect in osteoblasts following 7 days of incubation (*p* < 0.05 compared to the negative control). 

In the case of HAp@Zn nanoparticles, the reduction in cells’ metabolism was more pronounced at all of the investigated concentrations compared to HAp and HAp@B samples. This inhibitory effect is probably induced by the presence of Zn in the composition of the nanoparticles, which is a well-known effect determined by Zn^2+^ ion dissolution [53,54].

The addition of gypsum showed an improved effect on osteoblast cell viability at 25 μg/mL (HAp@Zn vs. G+HAp@Zn, *p* < 0.01). 

Differentiation of osteoblast cells involves the mineralisation of the extracellular matrix in order to form bone matrix, similar to the in vivo environment [55]. The Ca deposits were evidenced in the nanoparticle-treated osteoblasts using a specific reaction between alizarin red and the mineralised areas in the extracellular matrix of the cells (Figure 15). Using this test, the native ability of HAp alone to induce differentiation of osteoblast cells at 7 days of incubation was clearly evidenced. For all samples, blank samples were prepared at equivalent concentrations in order to remove the possible interferences induced by the presence of nanoparticles alone. Making a correlation with MTT assay data, when differentiation occurs in osteoblast cells, proliferation is usually inhibited, and a reduction in the cells’ metabolism takes place [56,57]. Thus, the reduction in metabolic activity of MG-63 exposed to HAp nanoparticles is correlated with differentiation data, where a significant amount of Ca deposits was measured. 

Although gypsum alone inhibited the differentiation of osteoblasts in terms of calcium deposition, this suppressing effect did not exceed 30% for all concentrations (*p* < 0.05). In consequence, this effect was also translated to cells exposed to gypsum-modified HAp, where a decrease in Ca deposit production was measured compared to control cells. However, this effect was statistically significant only for 200 μg/mL concentration. 

The addition of B to the composition of HAp nanoparticles clearly improved the ability of osteoblast-like cells to differentiate and mineralise the extracellular matrix in vitro. The amount of Ca deposits was significantly higher compared to negative control samples at all concentrations. The high ability of HAp@B and G+HAp@B to induce mineralisation in MG-63 cell culture was correlated with their metabolism inhibition. 

A similar effect of differentiation improvement was measured in the cases of HAp@Zn and G+HAp@Zn compared to the negative control. However, the addition of gypsum to the composition of HAp@Zn samples did not increase the amount of resulting calcium deposits as compared to HAp@Zn alone. Similarly, the differentiation of osteoblasts exposed to HAp@Zn and G+HAp@Zn was correlated with a reduction in the cells’ metabolism compared to the negative control.

## 4. Conclusions

The study presents the results of the preparation and characterisation of the composite obtained by mixing gypsum with Zn- or B-doped hydroxyapatite nanoparticles for hard tissue restoration.

The experiments showed that the addition of the dopant led to better crystallisation. By adding 2% Zn, a 2.27% increase in crystallinity was observed, while the addition of B led to an increase of 5.61% compared to HAp. The crystallite size decreases as a result of doping. For B, a crystallite size of 10.69 ± 1.59 nm was measured, and in the case of Zn, the size value was 16.63 ± 1.83 nm, compared to HAp, where the crystallite size value was 19.44 ± 3.13 nm.

In the case of the HAp, medium-sized particles are observed around the value of 67.11 nm, while the values decrease with the addition of dopants, reaching 54.67 nm for the sample doped with B and 52.64 nm for the sample doped with Zn. Also, due to their small size and pseudo-acicular shape, the samples show an accentuated agglomeration tendency.

After hardening, the composites, obtained by mixing the gypsum with the HAp-type particles and the aqueous solution, were matured for 28 days and characterised by FTIR. 

The morphology of the composite samples is similar and is predominantly formed by the characteristic forms of calcium sulphate dihydrate. Samples are formed by crystals with sizes between 200–350 nm. HAp particles are uniformly distributed on the surface of the gypsum crystals due to the mechanical homogenisation process.

The rate of degradation of the composites in the SBF environment is accelerated in the first 15 h when a decrease in the weight of the samples by approximately 37% is recorded. After a decrease in degradation rate is observed, the sample reaches a maximum of approximately 47% after 72 h of contact with the liquid. This rate of disintegration of the resulting paste may coincide with the beginning of cell proliferation to form new bone tissue.

It is also noted that G+HAp@Zn samples tend to disintegrate more slowly in the presence of SBF compared to the other types of studied composites that showed similar behaviour. After the introduction of samples in SBF, these resistances decreased by about 6%. 

The ability of HAp, HAp@B, and HAp@Zn nanoparticles, respectively, to mineralise MG-63 osteoblast-like cells was clearly highlighted at all concentrations involved in the study. This effect is correlated with the reduction in the cells’ metabolism following the differentiation process, which normally takes place when the mineralisation of bone cells occurs.

## Figures and Tables

**Figure 1 nanomaterials-13-02219-f001:**
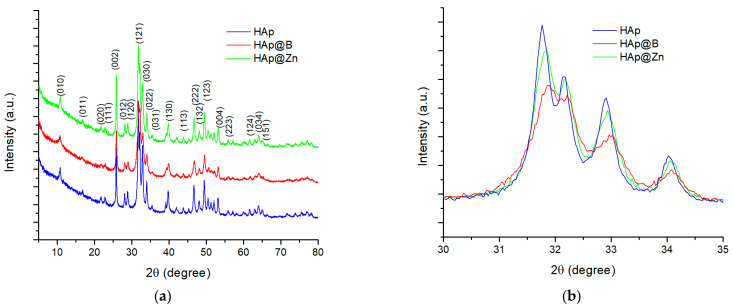
XRD full patterns for HAp (doped and undoped) powders (**a**) and detailed XRD patterns for the most intense peaks (**b**).

**Figure 2 nanomaterials-13-02219-f002:**
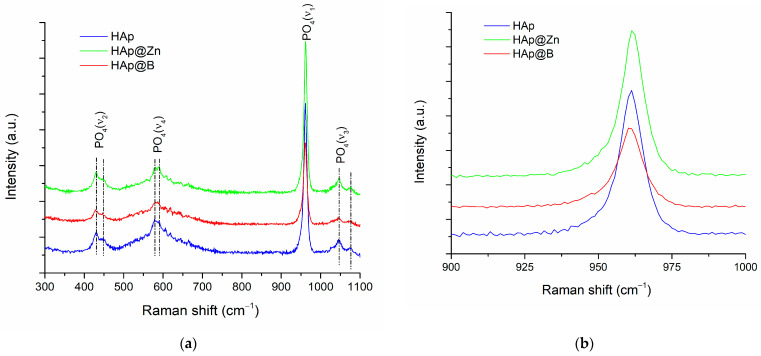
Raman spectra obtained for the studied samples (**a**) and representation of the most intense peaks (**b**).

**Figure 3 nanomaterials-13-02219-f003:**
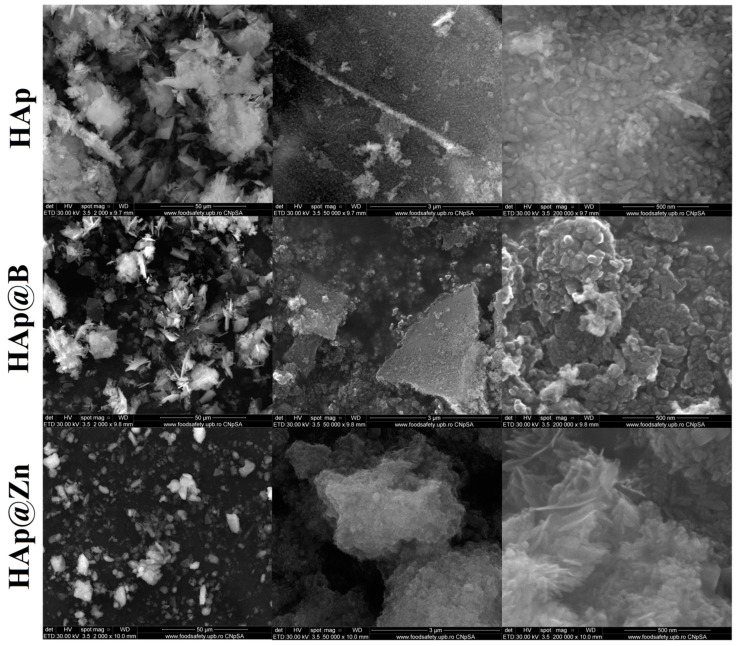
SEM images of the studied samples at different sizes (×2000, ×50,000, ×200,000).

**Figure 4 nanomaterials-13-02219-f004:**
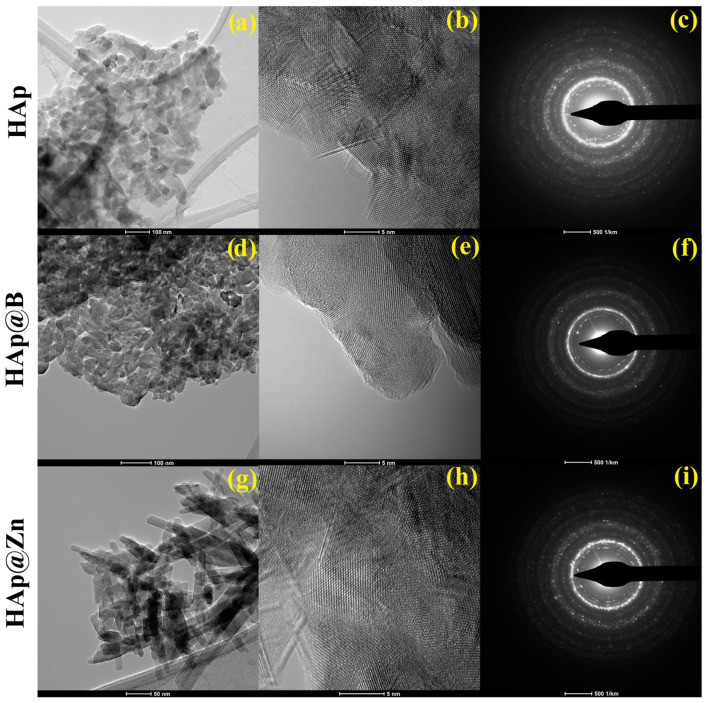
TEM (**a**,**d**,**g**), HRTEM (**b**,**e**,**h**), and SAED (**c**,**f**,**i**) analysis for the studied samples: HAp, HAp@B, and HAp@Zn.

**Figure 5 nanomaterials-13-02219-f005:**
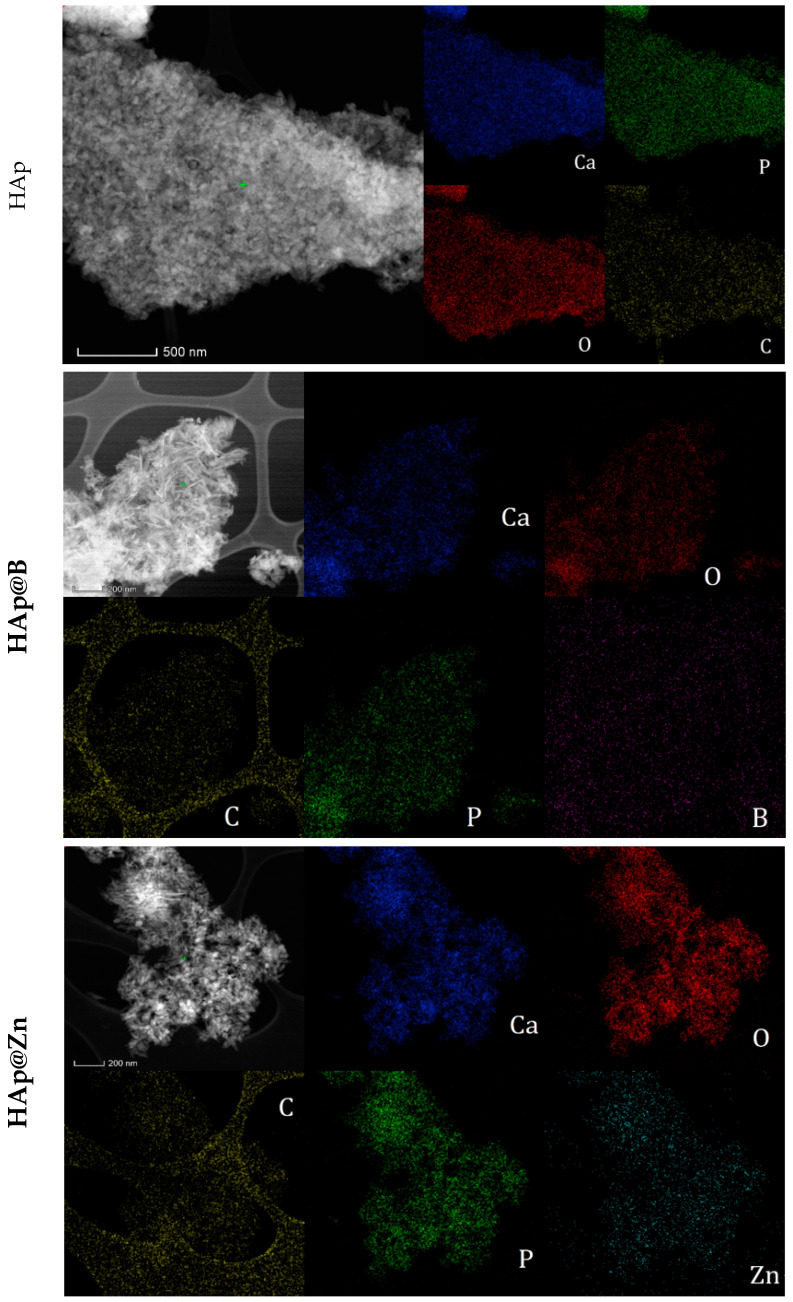
TEM mapping of elements on HAp, HAp@Zn, and HAp@B.

**Figure 6 nanomaterials-13-02219-f006:**
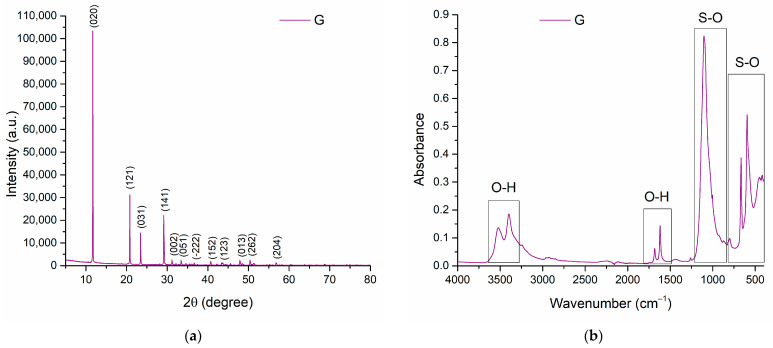
XRD diffractogram (**a**) and FTIR analysis (**b**) for gypsum powder.

**Figure 7 nanomaterials-13-02219-f007:**
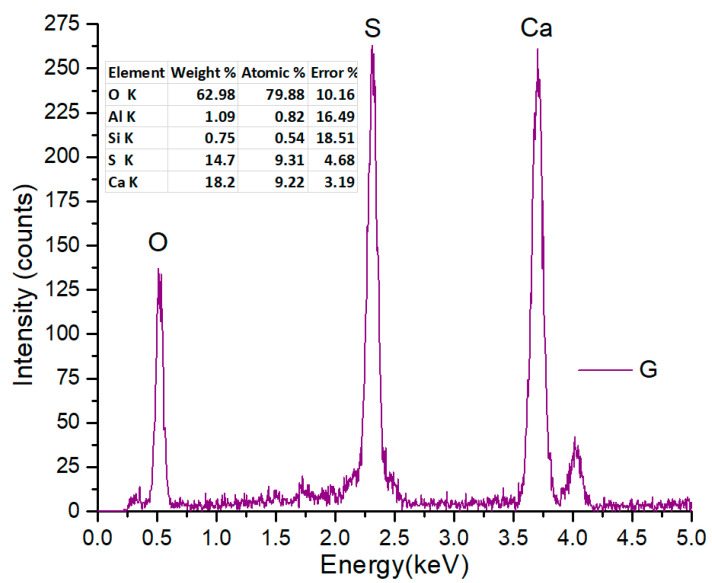
The EDS spectrum of gypsum and the corresponding composition.

**Figure 8 nanomaterials-13-02219-f008:**
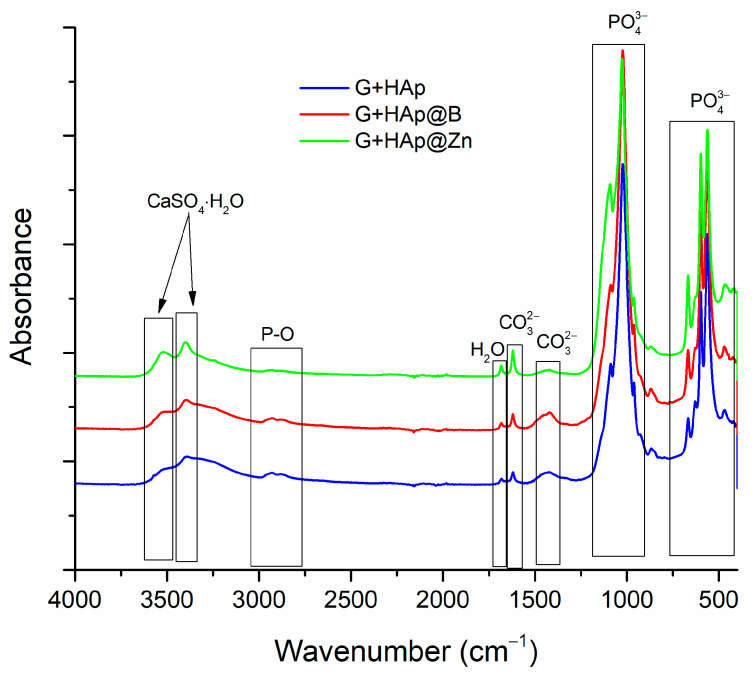
FTIR analysis was performed on the samples after 28 days.

**Figure 9 nanomaterials-13-02219-f009:**
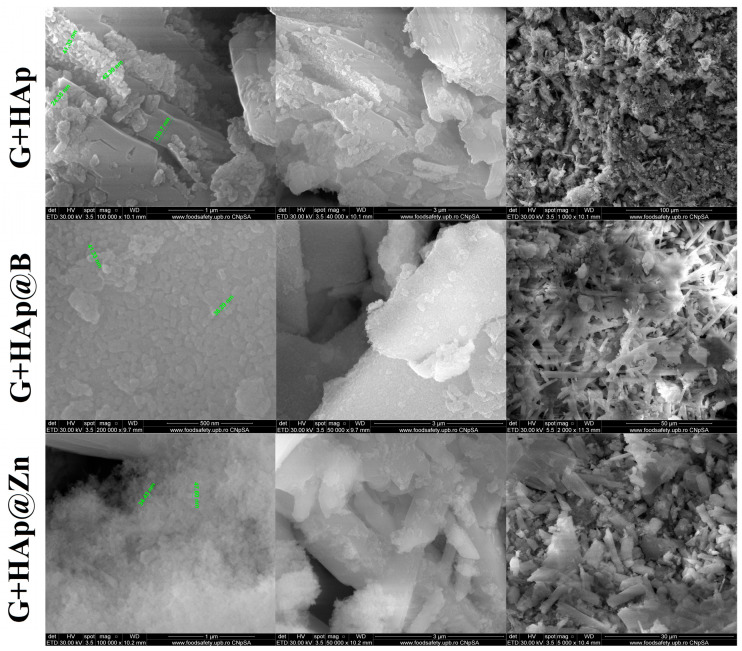
SEM images of the three studied composites.

**Figure 10 nanomaterials-13-02219-f010:**
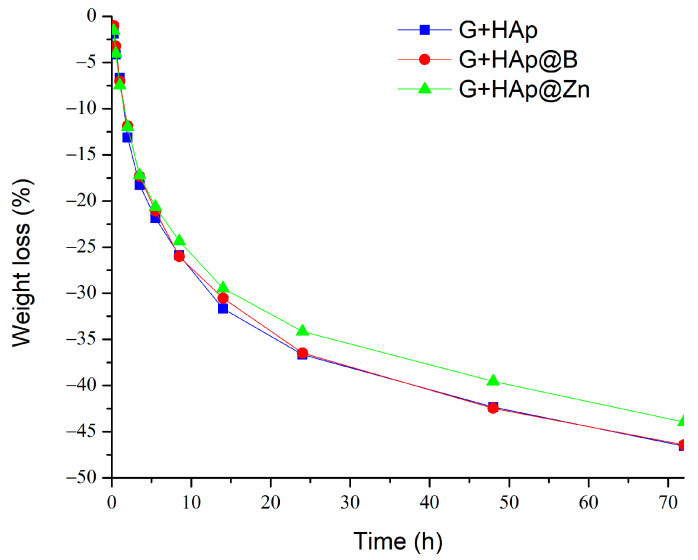
Weight loss of samples immersed in SBF for 72 h.

**Figure 11 nanomaterials-13-02219-f011:**
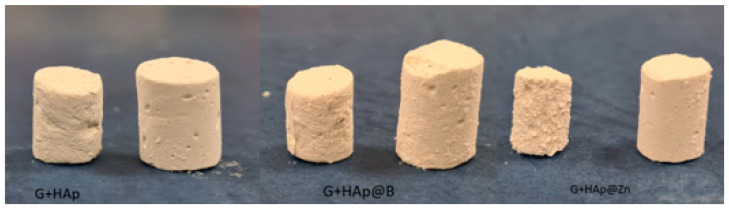
Images of samples after immersion in SBF (samples on the left) and non-immersed samples (samples on the right).

**Figure 12 nanomaterials-13-02219-f012:**
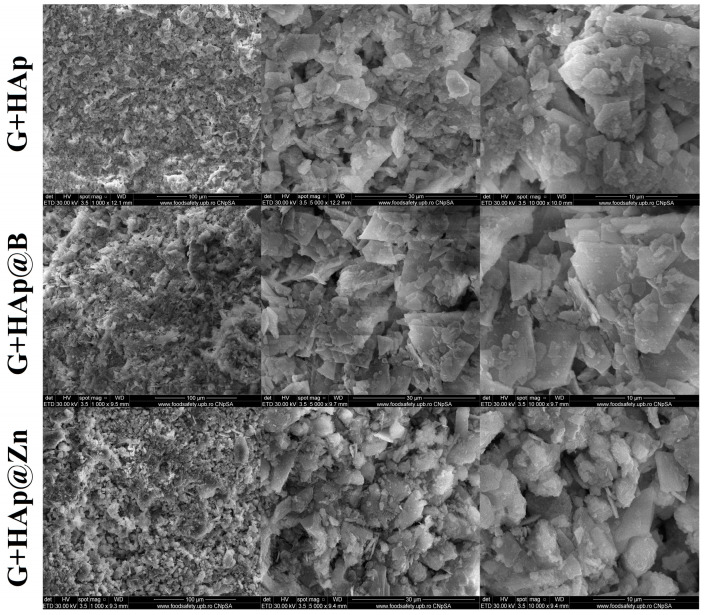
SEM images of samples at different magnifications (×1000, ×5000, ×10,000).

**Figure 13 nanomaterials-13-02219-f013:**
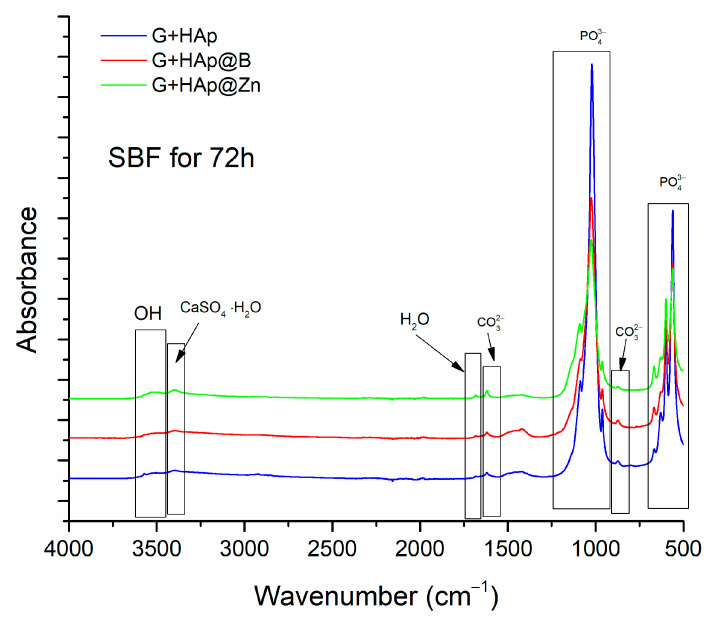
FTIR spectrum of the tested samples after 72 h immersion in SBF.

**Figure 14 nanomaterials-13-02219-f014:**
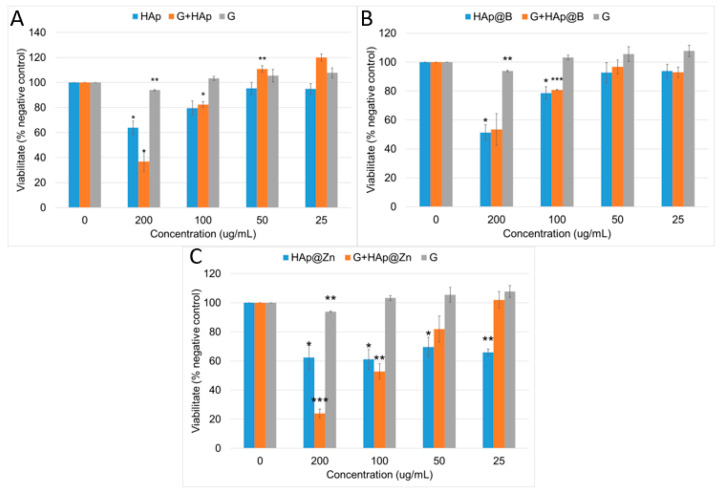
Viability and proliferation of MG-63 osteoblast-like cells incubated in the presence of gypsum/hydroxyapatite (**A**), gypsum/hydroxyapatite doped with B (**B**) and gypsum/hydroxyapatite doped with Zn (**C**) for 7 days; viability was calculated as reported to control cells (100%). Data were expressed as ±SEM, where * *p* < 0.05, ** *p* < 0.01, *** *p* < 0.001.

**Figure 15 nanomaterials-13-02219-f015:**
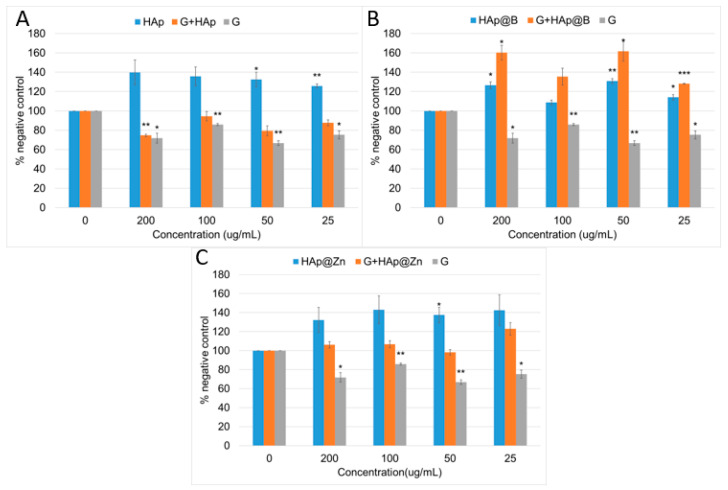
Differentiation of MG-63 osteoblast-like cells incubated in the presence of gypsum/hydroxyapatite (**A**), gypsum/hydroxyapatite doped with B (**B**) and gypsum/hydroxyapatite doped with Zn (**C**) for 7 days using Alizarin red assay. Data were expressed as a percent of negative control (100%) and ± SEM, where * *p* < 0.05, ** *p* < 0.01, *** *p* < 0.001.

**Table 1 nanomaterials-13-02219-t001:** Initial and final pH values of hydroxyapatite-based powders.

Sample	Initial pH	Final pH
HAp	5.11	9.25
HAp@B	4.92	9.15
HAp@Zn	4.87	9.62

**Table 2 nanomaterials-13-02219-t002:** The composition of the obtained composites.

Notation	Gypsum (%)	HAp (%)	HAp@B (%)	HAp@Zn (%)	Gly (%)	H_2_O (%)
G+HAp	26.60	26.60	-	-	4.68	42.12
G+HAp@B	27.18	-	27.18	-	4.56	41.08
G+HAp@Zn	27.18	-	-	27.18	4.56	41.08
G	62.50	-	-	-	3.75	33.75

**Table 3 nanomaterials-13-02219-t003:** Cell parameters provided by XRD-Rietveld analysis.

Elementary Cell Parameters	Sample		
	HAp	HAp@Zn	HAp@B
a (Å)	9.4240 ± 0.0008	9.4245 ± 0.0011	9.4116 ± 0.0019
b (Å)	9.4240 ± 0.0008	9.4245 ± 0.0011	9.4116 ± 0.0020
c (Å)	6.8851 ± 0.0007	6.8844 ± 0.0009	6.8834 ± 0.0015
V (Å^3^)	529.56	529.57	528.04
Crystallinity (%)	31.11	33.38	36.72
Average crystallite size (nm)	19.44 ± 3.13	16.63 ± 1.83	10.69 ± 1.59
Microstrain (%)	0.47	0.55	0.86

**Table 4 nanomaterials-13-02219-t004:** EDS composition of the studied samples.

Elements	HAp		HAp@B		HAp@Zn	
	Mass %	Error %	Mass %	Error %	Mass %	Error %
C K	16.37	9.96	5.51	15.91	3.91	16.03
O K	53.74	9.82	50.41	10.07	57.15	9.75
P K	10.80	4.40	13.32	4.63	14.29	4.67
Ca K	19.09	1.41	27.04	1.66	24.24	1.77
Zn K	-	-	-	-	0.33	18.59
B K	-	-	3.72	34.24	-	-

**Table 5 nanomaterials-13-02219-t005:** Mechanical resistance (in MPa).

	3 Days	7 Days	28 Days	SBF
Gypsum (G)	-	8.5	10.43	-
G+HAp	0.46	1.04	1.15	1.08
G+HAp@B	0.63	0.69	0.83	0.74
G+HAp@Zn	1.16	1.91	2.06	1.93

## Data Availability

Not applicable.

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
