# Peer review of "Fabrication and Characterisation of Calcium Sulphate Hemihydrate Enhanced with Zn- or B-Doped Hydroxyapatite Nanoparticles for Hard Tissue Restoration"

_nanomaterials, 2023, doi:10.3390/nano13152219_

Round 1

Reviewer 1 Report

The research presents an interesting methodology/procedure for obtaining materials based on calcium sulphate/phosphate phases by functionalisation with Zn and B with potential biomedical applications. The manuscript is very well presented, the methodology and characterisation of the samples is appropriate and the results and discussion are adequately explained.

However, the reviewer raises the following issues for consideration:

The introduction correctly presents the background and the framework of the research and its objectives. However, some comments on possible applications and biomedical implications of the developed composite material are missing (preferably at the end of the introduction).

Line 125-126. It may be interesting to indicate the pH values of the SBF solution.

Line 131. Composite preparation. The concentrations of the solutions are not reported.

Line 156. The dimensions of the mould cylinders obtained are not given.

Line 182. Was any drying method/step used after each SBF immersion?

Line 187. Indicate the equipment and parameters used for mechanical tests. Number of replicates.

Line 159-160. It is useful to clarify the methodology used to obtain the crystalline parameters: degree of crystallinity, the crystallite size, the crystallization type and to identify the phases. What do the authors mean by “crystallization type”? How has the degree of crystallinity been determined? Debye-Scherrer formula (Williamson-Hall / Warren-Averbach / or other) to determine the crystallite size?

Line 165. Parameters of the Rietveld refinement adjustments and software.

 Line 241.  Area

Table 4. Report the cell parameter data according to the Rietveld analysis.

Line 252. What term do the authors mean when they note "a better crystallization”?  

Section. HAp, HAp@Zn and HAp@B powder characterisation. Has CO3 been detected in the samples by Raman and FTIR as attributed by EDS and TEM mapping analysis? Could the incorporation of carbonate be related to changes in the crystalline characteristics of the precipitates?

Figure 6a. Avoid the use of the negative signs in Miller's indexes.

3.3. Composites material characterisation section. Did the authors characterise the composite samples by XRD? Could they report some of these results? Can the mechanical strength may be determined by some porosity parameter of the bulk sample generated by the gypsum dissolution processes? Could the Gyp/HAp phase differences be related to a higher relative proportion of apatite (and not new apatite) due to the dissolution of gypsum? The treatment was carried out at 132°C, could other semi-hydrated calcium sulphate phases (other than gypsum, for example bassanite) have formed under hydrothermal conditions?

Figure 8. Remove sample reference “28 zile”

Line 500. Substitute the term "paper" (e.g. study, research...).

Preferably use the symbols Zn and B throughout the manuscript.

Improve the resolution as much as possible in some of the figures.

Reviewer 2 Report

The manuscript “Fabrication and characterization of calcium sulphate hemihydrate enhanced with Zn or B doped hydroxyapatite nanoparticles for hard tissue restoration” was focused on the development of the composite based on calcium sulphate hemihydrate enhanced with Zn or B doped hydroxyapatite nanoparticles was fabricated and evaluated for use in bone graft applications. The article is well structured, well written and contains many interesting results and conclusions. However, in my opinion few features might be deepened and require minor revisions.

It would be advisable to describe the method of applicability of the described bone replacement material system, e.g. milling, 3D printing, bone cement, etc.

The release profile of different ions from the system would be absolutely necessary to assess their effect
